# Dynamic Formation of Green Tea Cream and the Identification of Key Components Using the “Knock-Out/Knock-In” Method

**DOI:** 10.3390/foods12162987

**Published:** 2023-08-08

**Authors:** Cheng Guo, Wangyang Shen, Weiping Jin, Xiwu Jia, Zhili Ji, Jinling Li, Bin Li

**Affiliations:** 1College of Food Science and Engineering, Wuhan Polytechnic University, Wuhan 430023, China; guocheng@whpu.edu.cn (C.G.); whwangyangshen@126.com (W.S.); jwpacademic@outlook.com (W.J.); jiaxiwu212@126.com (X.J.); jizhili@whpu.edu.cn (Z.J.); lijinling@whpu.edu.cn (J.L.); 2Key Laboratory for Deep Processing of Major Grain and Oil, Ministry of Education, Wuhan Polytechnic University, Wuhan 430023, China; 3Hubei Key Laboratory for Processing and Transformation of Agricultural Products, Wuhan Polytechnic University, Wuhan 430023, China; 4College of Food Science and Technology, Huazhong Agricultural University, Wuhan 430070, China; 5Key Laboratory of Environment Correlative Dietology, Ministry of Education, Huazhong Agricultural University, Wuhan 430070, China

**Keywords:** green tea cream, dynamic formation process, key components, colloid particles, “knock-out/knock-in” method

## Abstract

The composition of green tea cream is extremely complex, and identification of key components is a prerequisite for elucidating its microstructure formation mechanism. This study examined the dynamic changes in the content of components and properties of colloid particles during the formation process of tea cream by chemical analysis and dynamic laser scattering (DLS). A “knock-out/knock-in” method was developed and used to further explore the relationship between the interaction of these components and the microstructure formation of tea cream. The results revealed that polysaccharides, proteins, epigallocatechin gallate (EGCG), and caffeine were the main components involved in tea cream formation. These components participated in the formation process in the form of polysaccharide–protein and EGCG–caffeine colloidal particles. Consequently, there were synchronized dynamic changes in the levels of polysaccharides, proteins, EGCG, and caffeine. The “knock-out/knock-in” experiment revealed that the interactions between EGCG or caffeine and macro-molecule components were not the key factors in tea cream microstructure formation. However, it was found that the complexation between EGCG and caffeine played a crucial role in the formation of tea cream. The findings suggested that decreasing the concentrations of EGCG and caffeine could be useful in controlling tea cream formation during tea beverage processing and storage.

## 1. Introduction

Tea is a globally popular beverage that has become integrated into many people’s daily lives [1,2]. However, the production and storage of tea beverages is often difficult because of the complex storage properties of tea cream [3,4]. Tea cream refers to the spontaneous turbidity and precipitation phenomenon that affects the appearance and negatively influences the taste and nutritional value of tea beverages. It can even shorten the shelf-life of tea products. Unfortunately, a complete eradication of tea cream has not yet been achieved [5,6]. Therefore, addressing this technical obstacle requires in-depth research on the components, formation process, and mechanism of tea cream.

In the 1960s, Roberts [7] and Smith [8] were among the first to study the phenomenon of tea cream in black tea. Subsequently, Liang et al. [9] discovered the presence of tea cream in green tea beverages, where catechins and methylxanthines were identified as the main components. However, the composition of tea cream is extremely complex, and various covalent and non-covalent interactions occur during the tea cream formation process, making it difficult to understand its formation mechanism. Therefore, identification of key factors in the formation of tea cream from complex components is a prerequisite for elucidating its formation mechanism. After understanding this issue, simple models can be used to explain the multiple mechanisms in the complex process of tea cream microstructure formation.

Previous studies primarily focused on the final precipitate of tea cream, neglecting the initial and intermediate stages of its formation, which are crucial for inhibiting tea cream. Moreover, recent research has indicated that components in tea infusion can assemble to form nano-/micro-scale colloidal particles, and the process of tea cream formation can be understood as the coagulation of these colloidal particles from a physical chemistry perspective [10,11]. However, current studies have paid limited attention to the colloid characteristics of tea cream, resulting in a simplified and idealized research model that does not fully capture the complexity of actual tea infusion.

To elucidate the formation mechanism, this study recorded the changes in turbidity and tea cream content during the dynamic formation processes of green tea cream at 4 °C. The variation in catechins, caffeine, proteins, polysaccharides and metal elements (Mg, Mn, and Ca) during the early, middle, and final stages of tea cream formation was analyzed by chemical analysis with high-performance liquid chromatography (HPLC) and inductively coupled plasma–optical emission spectrometer (ICP-OES) to understand their respective trends. Furthermore, the distribution coefficient of each component was calculated to evaluate its affinity for tea cream. The study also investigated the dynamic changes in the properties of colloid particles by dynamic laser scattering (DLS) and laser Doppler velocimetry (LDV) during the formation process of green tea cream.

In this study, we adopted the “knock-out/knock-in” method, commonly used to identify pharmacodynamic substances in traditional Chinese medicine, to identify the key components of tea cream formation. Synergistic components in Chinese materia medica play a vital role in understanding the molecular basis of their overall therapeutic potential [12,13]. However, previous methods were unable to observe the interaction of multiple components in Chinese materia medica, making it difficult to determine the key active components [14,15]. To overcome these limitations, some scholars have proposed a method in which different components in Chinese materia medica are selectively removed from the whole, followed by their reintroduction at different doses, and the activity of each knock-out and knock-in sample is tracked. This method comprehensively considers the interrelationships between the whole and its component parts in Chinese materia medica [16]. Taking inspiration from this method, we applied the “knock-out/knock-in” approach to identify key components in the tea cream formation process. First, the small-molecule and the macro-molecule components in original tea infusion are separated through 72 h dialysis. Then, haze-active components are gradually added into the dialyzed tea infusion. By comparing the changes before and after the “knock-out” and “knock-in” experiments, the key components in the tea cream formation are able to be found, and the relationship between component interactions and tea cream formation can be further elucidated.

## 2. Materials and Methods

### 2.1. Materials

Epigallocatechin gallate (EGCG) (≥98%), epigallocatechin (EGC) (≥98%), epicatechin (EC) (≥98%), epicatechin gallate (ECG) (≥98%), and caffeine (≥98%) were purchased from Aladdin Biochemical Technology Co., Ltd. (Shanghai, China). The other analytical-grade reagents were purchased from Sinopharm Chemical Reagents Co., Ltd. (Shanghai, China). Milli-Q purified water producing by Milli-Q Direct (Merck, Germany) was used to conduct all experiments.

### 2.2. Preparation of Tea Infusion

Yunwu^®^ green tea (a non-fermented tea) was purchased from Yunwu Tea Industry Co., Ltd. (Wuhan, China). The ground tea leaves (20–60 mesh) were extracted with Milli-Q water at a ratio of 1:20 (*w*/*v*) and heated at 70 °C for 10 min, a magnetic stirrer (DF-101S, Lichen, China) was used to assist by stirring (120 r/min) throughout the extraction process [6]. The resulting tea infusion was centrifuged at 10,000× *g* for 15 min to obtain a clarified tea infusion, referred to as the original tea infusion.

### 2.3. Formation and Separation of Tea Cream

The original tea infusion was stored at 4 °C for 4 h to allow the formation of tea cream. The turbidity of the sample was monitored using a spectrophotometer (UV-1100, MAPADA, China) at a wavelength of 520 nm [17]. The infusion was centrifuged at 10,000× *g* for 15 min at 4 °C to separate the tea cream from the sample. After collecting the supernatant, the precipitated tea cream was washed with two 5 mL aliquots of Milli-Q water into a weighed glass dish. Subsequently, tea cream was dried at 80 °C for 48 h [18].

### 2.4. Analysis of Chemical Components

The concentrations of proteins and polysaccharides were determined using a UV-vis spectrophotometer (UV-1100, MAPADA, Shanghai, China), following the procedures described in previous reports [18,19]. The concentrations of catechins (EGCG, ECG, EGC, and EC) and caffeine were measured using an Agilent 1200 series HPLC system equipped with an Eclipse XDB-C18 column (4.6 × 250 mm, 5 μm, Agilent), as previously described [6]. The detection of Mg^2+^, Mn^2+^, and Ca^2+^ was performed using ICP-OES (720-ES, Agilent, Santa Clara, CA, USA) with an RF power of 1.2 kW [10].

### 2.5. SDS-PAGE Electrophoresis

The aqueous solutions of tea cream and tea infusion proteins were mixed with 5× electrophoresis sample buffer in a ratio of 4:1 (*v*/*v*). The mixture was then heated at 95 °C for 10 min. Subsequently, 10 μL of each sample was loaded onto the gel. The resolving and stacking gels had acrylamide concentrations of 12% and 5%, respectively. Protein staining and gel destaining methods were carried out as described in a previous report [20]. The proteins from tea infusion were extracted using an alkaline method which has been widely used for protein extraction from plant sources. Briefly, green tea leaves were mixed with a sodium hydroxide solution (0.08 mol/L) and stirred (120 r/min) continuously at 70 °C for 15 min. The extracting solution was then centrifuged at 1500× *g* for 10 min to collect the precipitated proteins.

### 2.6. Measurements of Colloid Particles Properties in Tea Infusion

The *z*-average hydrodynamic diameter (D*_z_*), polydispersity index (PDI), and zetapotential of colloid particles in tea infusion were measured using the Zetasizer Nano-ZS90 instrument (Malvern Instruments Ltd., Malvern, UK) [6].

### 2.7. Calculation of Distribution Coefficient

The distribution coefficient (*K*_A_) of each component between the tea cream and the supernatant was calculated with slight adjustments according to the method of Penders et al. [21].
$$K_{A}=\frac{mass\;of\;A\;in\;tea\;cream}{mass\;of\;A\;in\;supernatant}$$

### 2.8. The “Knock-Out/Knock-In” Method

#### 2.8.1. “Knock-Out” of Small-Molecule Components

The small-molecule components present in the original tea infusion were removed through a 72 h dialysis process using 3500 Da molecular weight cutoff tubing. The dialysis was performed against Milli-Q water at room temperature [6]. After dialysis, the tea infusion was allowed to stand for 4 h at 4 °C. Subsequently, measurements were taken for D*_z_*, PDI, and zeta potential.

#### 2.8.2. “Knock-In” of Key Components

After 72 h dialysis, the small-molecule components present in the original tea infusion were knocked out, and the remaining portion (referred to as the dialyzed tea infusion) was used for the next “knock-in” experiment. Different amounts of EGCG and caffeine were added into six bottles of dialyzed tea infusion (Bottle 1: EGCG 0 mg/mL, caffeine 0 mg/mL; Bottle 2: EGCG 0 mg/mL, caffeine 1.0 mg/mL; Bottle 3: EGCG 1.0 mg/mL, caffeine 0 mg/mL; Bottle 4: EGCG 1.0 mg/mL, caffeine 0.5 mg/mL; Bottle 5: EGCG 1.0 mg/mL, caffeine 1.0 mg/mL; Bottle 6: EGCG 1.0 mg/mL, caffeine 1.5 mg/mL). All the samples were allowed to stand for 4 h at 4 °C. Following this, the changes in turbidity were analyzed.

### 2.9. Statistical Analysis

SPSS statistical software V22.0 (IBM, Armonk, NY, USA) was used to perform statistical analysis. In the chemical component analysis and “knock-out/knock-in” experiment, the mean and standard deviation of five replicates were calculated. The measurement of size distribution and the detection of metal elements were performed three replicates. The statistically significant difference was tested by one-way ANOVA, and this was followed by a least significant difference test. Differences were considered to be statistically significant when *p* was <0.05.

## 3. Results and Discussion

### 3.1. Formation Process of Green Tea Cream at 4 °C

Tea cream spontaneously forms in the tea infusion during the cooling process. In our experiment, in order to avoid the influence of microbial activity, the original tea infusion was stored at a cold temperature (4 °C). Additionally, tea cream will quickly form at 4 °C, and the interactions among the components in tea infusion are more obvious. As depicted in Figure 1A, visible hazes gradually appeared in the samples with the prolongation of storage time, becoming more pronounced after 1 h of storage at 4 °C. The changes in turbidity of the tea infusion during storage at 4 °C are presented in Figure 1B. The turbidity sharply increased from 30.80 ± 2.48% to 82.70 ± 4.24% between 0.5 and 3 h, and the rate of turbidity growth leveled off after 3 h of storage. Simultaneously, the content of tea cream also increased from 1.55 ± 0.35 mg/mL to 3.95 ± 0.21 mg/mL and reached a plateau after 4 h (Figure 1C). These findings revealed that the turbidity and the tea cream content both stopped increasing after 4 h, indicating tea cream was no longer formed. Therefore, tea cream has been developed fully after 4 h of storage at 4 °C. Previous studies have reported a significant effect of temperature on tea cream formation, with lower temperatures leading to faster cream formation [11,22,23]. Tea cream primarily consists of tea polyphenols, methylxanthines, proteins, and polysaccharides. These components exist in the hot tea infusion in a free state. However, as the temperature decreases, intermolecular interactions strengthen, causing the components to combine and form tea cream [24,25,26]. Consequently, the interaction among the components in the tea infusion strengthens at 4 °C, resulting in the rapid formation of tea cream.

### 3.2. Dynamic Changes in Chemical Component Contents Observed during the Green Tea Cream Formation Process

Based on the changes in turbidity and tea cream contents, the formation process of green tea cream can be divided into three stages: initial, developmental, and fully formed. Therefore, we selected samples at 0.5, 2, and 4 h for analysis. To ensure accurate results, we considered that the combination of components in tea cream could interfere with the chemical analysis. Thus, we subtracted the component content in the supernatant after removing the precipitate from the component content in the original tea infusion. This approach allowed us to obtain precise and reliable results.

Figure 2A illustrates the dynamic changes in the main components in green tea cream. Polysaccharides were found to be the component with the highest content in tea cream. The polysaccharide content increased rapidly from 163.17 ± 39.85 mg/mL to 362.38 ± 55.29 mg/mL between 2 and 4 h, indicating a significant involvement of polysaccharides in the later stage of tea cream formation. Moreover, the protein content in tea cream also began to increase rapidly after 2 h of storage at 4 °C. This suggests that the dynamic changes in polysaccharides and proteins were synchronized, possibly due to their simultaneous participation in tea cream formation. Previous studies have reported the presence of green tea colloid particles, which are polysaccharide–protein conjugates known to play a crucial role in tea cream formation [6,11]. Therefore, when polysaccharides and proteins participate in tea cream formation in the form of colloid particles, the content changes in these two components remain synchronized.

The molecular weights of proteins present in tea cream were further determined and depicted in Figure 3. The proteins in tea cream were found to have molecular weights of 10, 15, and 15–25 kDa. Proteins with molecular weights above 25 kDa in the tea infusion did not participate in tea cream formation. This observation suggests that proteins with molecular weights of 10, 15, and 15–25 kDa could form tea creams. Furthermore, it is noteworthy that although the content of proteins in tea cream increased with the standing time, the molecular weights of the proteins in tea cream remained unchanged. This phenomenon indicates that the proteins responsible for the haze formation did not undergo molecular weight changes during the tea cream formation process.

EGCG and caffeine were identified as the other two main components in green tea cream. These two components have a strong affinity for each other and can immediately form a precipitate at low temperatures [6,27]. However, the contents of EGCG and caffeine decreased significantly from 2 to 4 h (Figure 2A). This decrease can be attributed to two factors. First, the increased contents of polysaccharides and proteins may have influenced the overall concentration of EGCG and caffeine. Second, the amount of precipitates formed by combining EGCG and caffeine in the late stage of tea cream formation decreased.

Additionally, Figure 2B shows a significant decrease in the contents of Mg^2+^, Mn^2+^, and Ca^2+^ in tea cream from 2 to 4 h. Metal ions, especially certain divalent ions, have been found to enhance tea cream formation by promoting the self-aggregation of EGCG and facilitating the hetero-aggregation between EGCG and caffeine [28,29]. The decreased contents of these divalent ions indicate that the combination of EGCG and caffeine was weakened in the late stage of tea cream formation.

### 3.3. Comparison of Distribution Coefficients of Chemical Components

It is important to note that the content of each component alone cannot fully reflect its affinity for participating in tea cream formation, as the original concentration of the component in the tea infusion can influence it. We introduced distribution coefficients, as described by Penders et al. [21], to evaluate the affinities of components for tea cream formation and address this issue. However, due to the extremely low contents of metal ions in tea cream, the distribution coefficients for metal ions were not calculated in this study. As shown in Figure 2C, the distribution coefficients of gallate-type catechins (EGCG and ECG) were higher than those of nongallate-type catechins (EC and EGC). This suggests that gallate-type catechins exhibit stronger aggregating affinities, while nongallate-type catechins tend to remain in the supernatant. This can be attributed to the galloyl group present in gallate-type catechins, which promotes their binding with caffeine [10,30]. Second, Figure 2C also demonstrates that the distribution coefficients of polysaccharides and proteins were not significantly different, as were the distribution coefficients of EGCG and caffeine. This provides further evidence that polysaccharide and protein, as well as EGCG and caffeine, synchronously participate in tea cream formation. In conclusion, considering both the contents and the distribution coefficients, EGCG, caffeine, polysaccharides, and proteins can be considered key components in tea cream formation.

### 3.4. Dynamic Changes in the Properties of Colloid Particles Associated with the Green Tea Cream Formation Process

In addition to the free chemical components, tea infusion contains nano-/micro-scale colloidal particles [29,31]. These colloidal particles include green tea nanoparticles, which are polysaccharide–protein conjugates considered to be precursors of tea cream formation [6,32,33,34,35]. The size distributions of the nano-/micro-scale colloidal particles in the supernatant of tea infusion were determined after removing the precipitate at different time points (0.5, 2, and 4 h at 4 °C) to assess the dynamic changes in colloid particle properties.

Figure 4 displays the size distributions of colloidal particles in the original tea infusion and the supernatant at different time points. In the size distribution of the original tea infusion, a single peak was observed, with a *z*-average hydrodynamic diameter (D*_z_*) of 464.1 ± 23.1 nm. However, as the standing time increased, the size distributions of colloidal particles in the supernatant revealed a small additional peak in the range of 40–120 nm. The D*_z_* value decreased to 331.5 ± 0.28 nm when the samples were allowed to stand for 4 h. These findings indicate that the colloidal particles in the original tea infusion continuously participate in tea cream formation during the standing process at 4 °C, resulting in the concurrent formation of smaller-sized particles. As previously described, EGCG and caffeine rapidly combine to form small-sized complexes, which subsequently grow and contribute to the formation of precipitates [6,36]. Therefore, there are EGCG–caffeine complexes in the newly formed smaller-sized particles.

### 3.5. Identification of Key Components in Tea Cream following the “Knock-Out/Knock-In” Method

Based on the previous results, it can be concluded that proteins, polysaccharides, EGCG, and caffeine are the key components involved in tea cream formation, with the participation of green tea colloidal particles. The small-molecule components, such as catechins and caffeine, were separated from the tea infusion through dialysis to investigate the role of these components further. This resulted in the dialyzed tea infusion consisting mainly of free proteins, polysaccharides, and green tea colloidal particles. EGCG and caffeine were then individually added to the dialyzed tea infusion to observe the effects on tea cream formation. By comparing the changes before and after the “knock-out” and “knock-in” experiments, we could assess the relationship between the interaction of these components and tea cream formation.

#### 3.5.1. Changes in Tea Infusion after “Knock-Out” Experiment

Figure 5A shows no significant differences in the D*_z_* and PDI values between the dialyzed and original tea infusion samples. However, the zeta potential of the dialyzed tea infusion was significantly lower than that of the original tea infusion. This indicates that the colloid particles in the dialyzed tea infusion had a reduced propensity for aggregation and precipitation. Furthermore, the D*_z_*, PDI, and zetapotential of the dialyzed tea infusion remained stable after standing for 4 h at 4 °C. This suggests that tea cream formation would not occur solely due to the interaction among these macro-molecule components. The findings indicate that the interaction between proteins, polysaccharides, EGCG, and caffeine, along with the presence of green tea colloidal particles, plays a crucial role in tea cream formation.

#### 3.5.2. Changes in Tea Infusion after “Knock-In” Experiment

Different amounts of EGCG and caffeine were added to the dialyzed tea infusion samples to investigate further the relationship between small-molecule components and tea cream formation. All the samples were allowed to stand for 4 h at 4 °C, and the changes in turbidity were observed (Figure 5B). First, the dialyzed tea infusion remained clear when EGCG or caffeine was added separately. However, when EGCG and caffeine were added together, an obviously haze formed in the dialyzed tea infusion. This indicates that the interactions between EGCG or caffeine and the macro-molecule components alone were not the key factors in tea cream formation. Instead, the complexation between EGCG and caffeine played a crucial role. Gallate-type catechins, including EGCG, exhibit a strong affinity for caffeine. They can quickly bind together to form precipitates, especially at low temperatures [36,37]. Furthermore, as shown in Figure 5C, the EGCG–caffeine complexes can carry the nanoparticles and facilitate their precipitation through hydrophobic interactions [6]. In addition, Chen et al. reported that the colloidal particles of tea-polysaccharide conjugates were able to participate in the formation of EGCG–caffeine precipitation by promoting the interactions of EGCG and its C2 isomers [11]. In summary, the results of the “knock-out/knock-in” experiment provide an explanation for the synchronized dynamic changes observed in polysaccharides-proteins and EGCG–caffeine interactions. It confirms that the complexation between EGCG and caffeine is crucial in tea cream formation.

## 4. Conclusions

The formation mechanism of tea cream is highly complex and difficult to elucidate. In this study, we aimed to accurately identify the key components involved in green tea cream formation by analyzing the dynamic changes in chemical components and colloid particle properties. We also developed a “knock-out/knock-in” method to investigate the interaction between these components and tea cream formation. The results revealed that polysaccharides, proteins, EGCG, and caffeine were the main components involved in tea cream formation. These components were found to participate in the formation of tea cream as polysaccharide–protein and EGCG–caffeine colloidal particles. As a result, the dynamic changes in polysaccharides, proteins, EGCG, and caffeine were synchronized throughout the process. Furthermore, the “knock-out/knock-in” experiment demonstrated that the interactions between EGCG or caffeine and macro-molecule components were not key to tea cream formation. Instead, the complexation between EGCG and caffeine played a crucial role. Tea cream formation was driven by the complexation of EGCG and caffeine rather than the individual interactions between EGCG or caffeine and other macro-molecule components. This study provides valuable insights into the key components involved in tea cream formation. It emphasizes the importance of considering multiple mechanisms for explaining the complex process of tea cream formation. The findings suggest that decreasing the concentrations of EGCG and caffeine can be useful in controlling tea cream formation during tea beverage processing and storage. In addition, this study also demonstrates that the “knock-out/knock-in” method is able to quickly and accurately identify the key factors in complex interaction, and it is useful for the synergistic effect research of complex components in food system.

## Figures and Tables

**Figure 1 foods-12-02987-f001:**
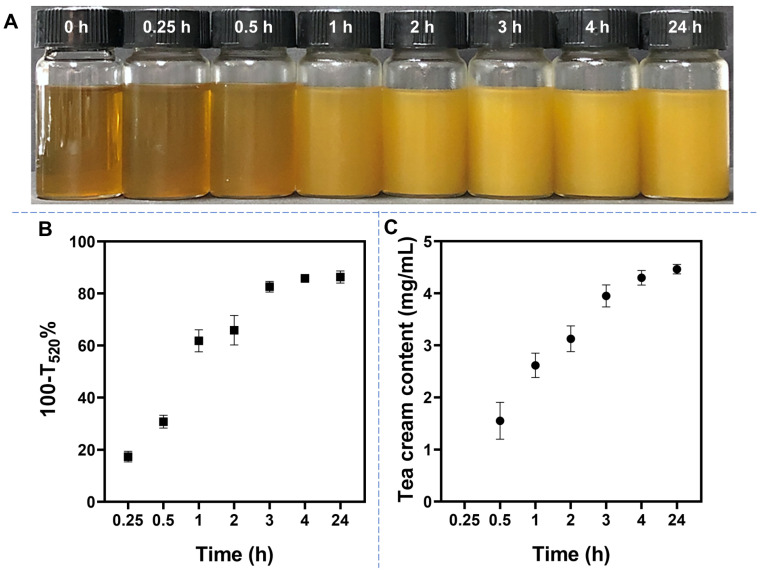
(**A**) Transformation of original tea infusion during storage at 4 °C. Changes in turbidity (**B**) and tea cream content (**C**) in original tea infusion stored at 4 °C. Data are shown as the mean ± SD, *n* = 3.

**Figure 2 foods-12-02987-f002:**
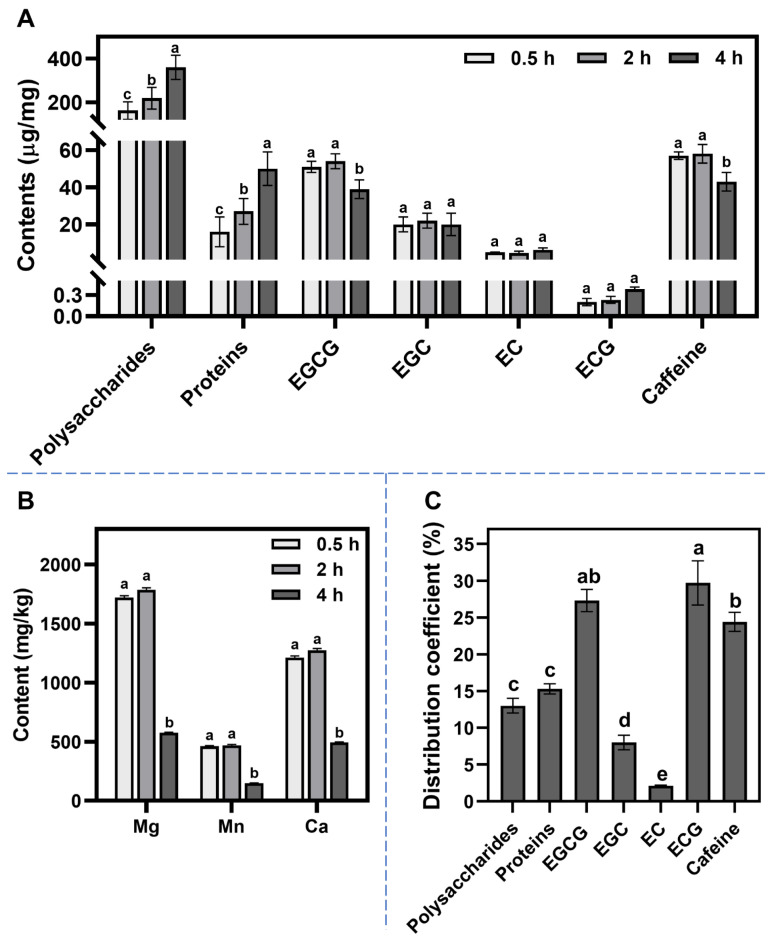
Changes in major chemical component contents (µg/mg) (**A**) and metal element contents (mg/kg) (**B**) during the green tea cream formation process. (**C**) Distribution coefficients of major chemical components of green tea cream stored at 4 °C for 4 h. Data are shown as the mean ± SD, *n* = 3. Different lower-case letters denote significant differences (*p* < 0.05) at each variable.

**Figure 3 foods-12-02987-f003:**
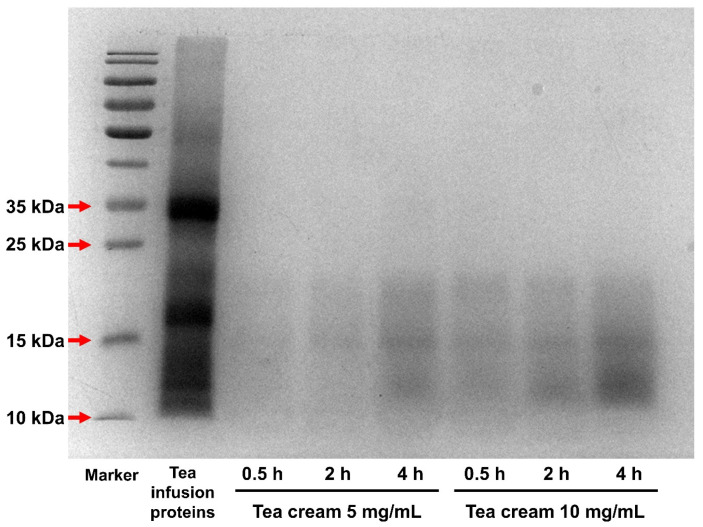
Electrophoretic pattern of proteins in green tea cream.

**Figure 4 foods-12-02987-f004:**
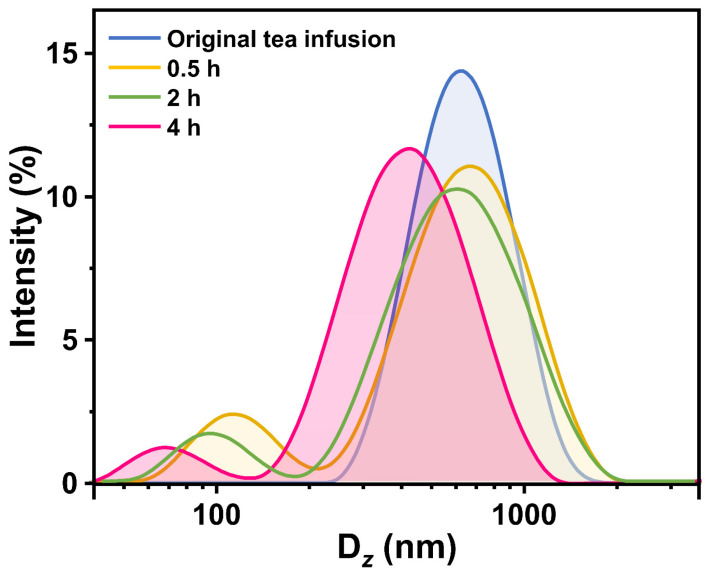
Changes in size distributions during green tea cream formation at 4 °C.

**Figure 5 foods-12-02987-f005:**
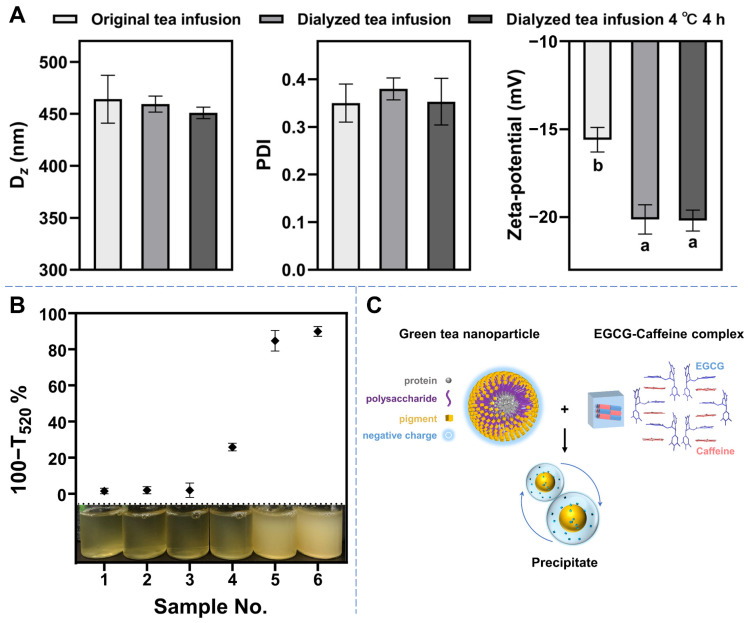
(**A**) Changes in the *z*-average hydrodynamic diameter, PDI, and zetapotential of dialyzed tea infusion recorded at 4 °C. (**B**) Changes in the turbidity of the samples recorded after adding EGCG and caffeine into dialyzed tea infusion at 4 °C. (**C**) Schematic illustration for the EGCG–caffeine-nanoparticle precipitation process. Data are presented as the mean ± SD, *n* = 3. Different lower-case letters denote significant differences (*p* < 0.05) at each variable.

## Data Availability

The datasets generated for this study are available on request from the corresponding author.

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
