# Peer review of "Dynamic Formation of Green Tea Cream and the Identification of Key Components Using the “Knock-Out/Knock-In” Method"

_foods, 2023, doi:10.3390/foods12162987_

Round 1

Reviewer 1 Report

The authors describe work to start to understand the development of tea cream when kombucha is cooled. They describe the experiment where they explore the composition of the tea cream and investigate the potential key components of caffeine and EGCG to the formation of the tea cream through a knock-out/knock-in type experiment. Overall, the paper is well written and clear. There are a couple of points which can be clarified. First while it is pretty clear the steps that were taken because of the way it is described it is somewhat unclear the difference between the two major formations of tea cream, with and without knock-out. For example, was only one tea from which a portion was allowed to form tea cream, and another was knocked out? Some clarity of this description would help. In addition, what was the sample sized used? Is there a dependence on sample size? And finally, the paper ends with the discussion of the knock-in results, stating the increase in caffeine and EGCG concentration resulted in more and faster tea cream formation. However, little to no data is presented to support this. This seems to be a major finding with little support and should be improved.

Overall, the English is appropriate and clear.

Reviewer 2 Report

Dear author, 

The article entitled 'Dynamic formation of green tea cream and the identification of key components using the "knock-out/knock-in" methoddiscusses an interesting topic. However, this article needs significant improvement due to several issues, including a bad story flow. My specific comments are:
1. Abstract: Please add (a) the importance of this research; (b) the significance of the finding; (c) Some key analysis used in this study
2. Abstract: What does the author mean with formation? Microstructure formation? Flavour formation?
3.  Introduction: (a) Does it determine the shelf-life of the product? (b) Is it possible that tea cream is formed because of microbial activity?
4. Introduction: Why is it important to study the complexity of tea cream? What is the goal/future perspective after understanding this issue?
5.  Introduction: Why  did the author choose cold temperature for storage? 6. Introduction: Please explain the key principle of knock-out/knock-in method. 
7. Section 2.2: (a) Green tea leaves? How is the criteria of the tea leaves? In fact tea leaves can be differ by its enzymatic oxidation process determining its quality and key components
8. Section 2.2: How did the author do estraction? Simple mixing? Assisted by stirring (for how long)?
9. Line 146: Please provide the reason of your statement (term of "developed fully). Based on the statistical analysis?
10. Line 237: Why do the author only focus in nanoparticles? The distribution of nano-/micro-particles changes. Please also refer to the consensus of the definition of nanoparticles. 
11. Line 247-248: Can the author prove the EGCG-caffeine complexes?
12. Section 3.5: The data must be divided into two: (a) Sample after knock out method (compared to the control) and (b) Sample after knock in method (compared to the control).
13. Conclusion: The conclusion must be followed by a suggestion regarding the the practical application for delaying tea cream formation
